# RBD Double Mutations of SARS-CoV-2 Strains Increase Transmissibility through Enhanced Interaction between RBD and ACE2 Receptor

**DOI:** 10.3390/v14010001

**Published:** 2021-12-21

**Authors:** Siddharth Sinha, Benjamin Tam, San Ming Wang

**Affiliations:** MOE Frontiers Science Center for Precision Oncology, Faculty of Health Sciences, University of Macau, Taipa, Macau 999087, China; siddharths@um.edu.mo (S.S.); benjamintam@um.edu.mo (B.T.)

**Keywords:** SARS-CoV-2, variants, md simulations, free energy, antibody escape, MM/GBSA

## Abstract

The COVID-19 pandemic, caused by SARS-CoV-2, has led to catastrophic damage for global human health. The initial step of SARS-CoV-2 infection is the binding of the receptor-binding domain (RBD) in its spike protein to the ACE2 receptor in the host cell membrane. Constant evolution of SARS-CoV-2 generates new mutations across its genome including the coding region for the RBD in the spike protein. In addition to the well-known single mutation in the RBD, the recent new mutation strains with an RBD “double mutation” are causing new outbreaks globally, as represented by the delta strain containing RBD L452R/T478K. Although it is considered that the increased transmissibility of double-mutated strains could be attributed to the altered interaction between the RBD and ACE2 receptor, the molecular details remain to be elucidated. Using the methods of molecular dynamics simulation, superimposed structural comparison, free binding energy estimation, and antibody escaping, we investigated the relationship between the ACE2 receptor and the RBD double mutants of L452R/T478K (delta), L452R/E484Q (kappa), and E484K/N501Y (beta, gamma). The results demonstrated that each of the three RBD double mutants altered the RBD structure and enhanced the binding of the mutated RBD to ACE2 receptor. Together with the mutations in other parts of the virus genome, the double mutations increase the transmissibility of SARS-CoV-2 to host cells.

## 1. Introduction

The SARS-CoV-2 pandemic has caused devastating consequences to global public health, with over 270 million people infected and over 5.3 million lives lost globally since the COVID-19 pandemic started (https://covid19.who.int, accessed 16 December 2021). SARS-CoV-2 infects human cells through its spike (S) protein. In the process, the receptor-binding domain (RBD, residues 318–526) of the S protein (residues 1–1273) binds to the angiotensin-converting enzyme 2 (ACE2) receptor on the host cell membrane to release its genome into the cell [1,2]. Therefore, the RBD is a key determinant for SARS-CoV-2 infection. As an RNA virus, the genome of SARS-CoV-2 is constantly evolving with new mutations generated across its genome, including the RBD [3]. Since the first SARS-CoV-2 genome sequences, reported on 5 January 2020, there have been 1023 coding-changing mutations identified within the 193 positions of the RBD, which equals 1.52 mutation per day (1023 mutations in 675 days), and 5.3 mutations per position on average (Appendix A, http://cov-glue.cvr.gla.ac.uk/#/replacement, accessed 11 November 2021). Most of the mutations do not show pathogenic significance. However, the strains containing several RBD mutations, namely, L452R (epsilon), T478K, E484K, E484Q, and N501Y, were selected (Figure 1) and have caused multiple outbreaks, mostly due to the increased transmissibility of SARS-CoV-2 caused by these RBD mutations [4], (https://www.who.int/en/activities/tracking-SARS-CoV-2-variants/, accessed 15 December 2021). Recently, several new SARS-CoV-2 strains with RBD “double mutations” are causing new challenges [5]. The double mutations basically contain the same single mutations noted above, but they are more transmissible than the strains with the single mutation. For example, the “delta” strain, with the RBD mutation L452R/T478K, rapidly spread globally since it was identified in late 2020 [6]. Thus, understanding the mechanism of increased transmissibility in RBD double mutations is urgently needed to develop strategies to control their spread. Although studies have revealed how RBD single mutations increased SARS-CoV-2 transmissibility, it remains to be determined whether SARS-CoV-2 with RBD double mutations has adopted the same or similar manners as SARS-CoV-2 with RBD single mutations, or gains new features considering the more aggressive nature of SARS-CoV-2 with RBD double mutations than SARS-CoV-2 with single RBD mutations [7].

In this study, we investigated the changes in binding pattern and structural conformation between the ACE2 receptor and three major RBD double mutants of L452R/T478K (delta), L452R/E484Q (kappa) and E484K/N501Y (beta, gamma). We used multiple computational methods to address the topic, including molecular dynamics simulation (MDS), superimposed structural comparison, free binding energy estimation, and antibody escaping. Molecular dynamics simulation measures conformational change of the RBD mutant across a time period and uses the trajectories to describe the thermodynamic changes in RBD mutant structure [8,9]; superimposed structural comparison allows direct visualization of the RBD mutant structure altered by the double-mutated residues [10]; free binding energy change estimates the affinity change caused by the double-mutated residues between the RBD and ACE2 receptor [11]; and structural mapping of the antibody binding site explains whether RBD structural changes caused by the mutated residues can result in the escape of mutants from neutralizing antibodies (NAbs). The results from our study provided evidence that the three RBD double mutants altered the RBD structure in the ways much different from those caused by the corresponding RBD single mutants, enhanced the binding of the mutated RBD to ACE2 receptor, and can contribute to the increased transmissibility of SARS-CoV-2 to the host cells.

## 2. Materials and Methods

### 2.1. Structures of RBD and Mutated RBDs

The structure of the SARS-CoV-2 RBD–ACE2 receptor complex (PDB ID: 6M0J) was used as the wild-type reference [1]. The structure was optimized using a CHARMM36m force field in GROMACS version 5.1.2 (University of Groningen, Groningen, Netherlands). Three RBD double mutants, L452R/T478K (delta), L452R/E484Q (Kappa) and E484K/N501Y (beta, gamma), and five RBD single mutants of L452R, T478K, E484Q, E484K, and N501Y, included in the double mutants, were used in the study. The structures with the mutated residues in each mutant were generated using UCSF Chimera (RBVI, University of California, California, SF, United States) following the default parameters [12]. We compared the mutated N501Y, L452R, and T478K structures with other known SARS-CoV-2 crystal structures (PDB ID: 7MJN, 7ORB and 7ORA), and observed a 99.48%, 99.46%, and 99.46% similarity, respectively, indicating that the mutated crystal structures were comparable to the reference crystal structure.

### 2.2. Molecular Dynamics Simulation

GROMACS v.5.1.2 was used for the molecular dynamics simulation [13]. The CHARMM36m force field was chosen to model the RBD–ACE2 receptor complex [14]. The complex was situated in the simulation box 2 nm away from the box edge. The system was solvated with tip3p water and neutralized with Na+ ions. A steep descent algorithm was applied to the system before 1 ns equilibration, run at 298 K and 1 bar in the NPT ensemble, using a Berendsen thermostat and barostat. The system was set at 298 K and 1 bar in the NPT ensemble by using a V-rescale thermostat and a Parrinello–Rahman barostat during the production run [15]. A verlet velocity algorithm was employed with a time step of 2 fs. The Particle Mesh Ewald (PME) method was used to treat the long-range electrostatic interactions with the cut-off distance at 1.0 nm. The hydrogen bond was constrained at equilibrium lengths by the LINC algorithm and the trajectory frame of MD was saved every 30 ps [16]. RMSD, RMSF, Rg, SASA, and MSD were used to analyze structural changes of the native RBD and its mutants. The 35–40 ns simulation trajectories were utilized in each method. The XMGRACE program was utilized to generate the corresponding plots [17]. Three independent RMSD MD simulations were performed for the wild-type RBD–ACE2 for 100 ns and received RBD values at 0.186 nm, 0.173 nm, 0.179 nm, ACE2 values at 0.229 nm, 0.230 nm, 0.239 nm for run 1, run 2, run 3, respectively. The results confirmed that the RBD–ACE2 structure remained stable throughout the simulation process.

### 2.3. Superimposed Structural Comparison

Mutant RBD structures extracted at 100 ns of MD simulations through UCSF CHIMERA were superimposed with the wild-type RBD structures to identify the conformation changes using PYMOL.

### 2.4. Antibody Escape Analysis

To determine whether RBD mutants escape the binding by the neutralization antibody (NAb), structures between the mutant and wild-type RBD were structurally mapped with independent binding sites of the antibody C121 on RBD. The RBD binding site for antibody C121, a neutralizing antibody known to bind RBD (PDB ID: 7K8X) [18,19], and the E484K mutant known to escape C121 binding were used to determine the impact of the double mutants L452R/T478K, L452R/E484Q and E484K/N501Y on the antibody binding structure.

### 2.5. Free Binding Energy Calculation

MM/GBSA (molecular mechanics energies combined with generalized Born and surface area continuum solvation) [20] was used to calculate the free binding energy of the RBD to ACE2 (http://cadd.zju.edu.cn/hawkdock, Accessed 12 July 2021) [21]. The ff02 force field [22] and the GBOBC1 model [23] were assigned upon the proteins (Complex, RBD and ACE2) before the steepest descent and conjugate gradient minimization. The free energy of binding, Δ*G*, was calculated based on the following equation:(1)$$\Delta G=\;G\left(\mathrm{complex}\right)-\;G\left(\mathrm{RBD}\right)-\;G\left(\mathrm{ACE}2\right)$$

For each component, the total energy was estimated by using the following equation:


(2)
$$G_{\mathrm{bond}}\;=\;G_{\mathrm{elec}}\;+\;G_{\mathrm{vdW}}\;+\;G_{\mathrm{pol}}\;+\;G_{\mathrm{np}}\;-\;\mathrm{TS}$$


The first three terms represent the standard MM energy terms: bonded (bond, angle, and dihedral), electrostatic, and van der Waal interactions. G_pol and G_np are the polar and the non-polar contribution of solvation free energies, calculated by the Generalized Born (GB) implicit solvent method and solvent accessible surface area (SASA). The last term, absolute temperature, T, is multiplied by entropy, S. For the double mutant RBD and their single mutant RBD, the simulation was performed at 100 ns and a single protein coordinate file was extracted every 10 ns. Overall, free binding energy was averaged for each type of RBD.

## 3. Results

### 3.1. MD Analyses of Structural Changes in RBD Double Mutants

To investigate the effects of double mutations on the RBD structure, we examined the conformational changes of the RBD double mutant–ACE2 receptor complex in RBD double mutants L452R/T478K, L452R/E484Q, and E484K/N501Y using MD simulations for 100 ns. We performed the study by using five different types of method, including RMSD (root mean square deviation), RMSF (root mean square fluctuation), Rg (radius of gyration), SASA (solvent accessible surface area), and MSD (mean square displacement) for each type of RBD double mutant, with the wild-type RBD and five RBD single mutants as the controls. The results revealed substantial structural differences between the double mutants, wild-type, and single mutants:

**RMSD**: demonstrates the change in the atomic coordinates between the wild-type and mutant structures [24]. The wild-type RBD–ACE2 structure stabilized around ~0.3–0.4 nm with a deviation around 45 ns. L452R/T478K showed deviation up to ~30 ns and thereafter stabilized at ~0.3–0.4 nm; L452R/E484Q stabilized ~0.3–0.5 nm with a deviation at around 35 ns; E484K/N501Y reached the equilibrium distance at ~0.23 nm. Single mutants L452R, T478K, E484Q, and E484K, except for N501Y, showed stable configuration post 60 ns: L452R stabilized ~0.25 nm, T478K ~ 0.3 nm, E484Q ~ 0.2–0.3 nm, E484K ~ 0.3 nm. N501Y had 0.5 nm between 80 and 100 ns. The results showed that the trajectories in each of the RBD double mutants differed from the wild-type and the corresponding single mutants (Figure 2).

**RMSF**: determines the resilience of residues to analyze the effects of substitution in native and mutant structures [25]. The flexibility in the polypeptide chain through RMSF showed that double mutants L452R/E484Q, L452R/T478K, and E484K/N501Y had high to medium resilience ~ 0.4 nm, ~0.15 nm, and ~0.25 nm, respectively, at residue positions 360–380 and 470–490. In comparison, the single mutants had no greater flexibility in the backbone C-α atom than the native structure (Appendix A). The higher resilience in the double mutants L452R/E484Q and E484K/N501Y can be attributed to their differences in RMSD trajectory around residue positions 360–380 and 470–490.

**Rg:** measures the compactness between the wild-type and mutant structures [26]. The Rg value for the wild-type RBD was ~3.10–3.15 nm. The double mutant L452R/T478K showed a greater Rg value, ~3.2–3.25 nm, for the entire trajectory than the wild-type; L452R/E484Q had ~3.13–3.18 nm; E484K/N501Y had ~3.15–3.25 nm after 35 ns; L452R/E484Q and E484K/N501Y showed the change in Rg value ~80–100 ns and ~10–30 ns, respectively. Their larger hydrodynamic radius implies the change in the shape of the structure during protein folding and unfolding. T478K, E484K, and E484Q had higher Rg values than the wild-type RBD but lower than the double mutants, except for N501Y, which had a higher value, ~3.2–3.3 nm (Appendix A).

**SASA**: SASA defines the solvent-accessible surface area thereby measuring the relative expansion of the native and mutant structures [27]. The wild-type RBD had a surface area of ~430 nm^2^; all the single mutants, L452R, T478K, E484K, E484Q, and N501Y, had decreased values of ~426 nm^2^, ~426 nm^2^, ~424 nm^2^, ~428 nm^2^ and ~429 nm^2^, respectively. The double mutants also decreased values to these in single mutants, but their patterns were different (Appendix A).

**MSD**: MSD defines the mean square displacement of overall atoms from a set of initial positions between wild-type RBD and RBD mutants [28]. The wild-type of structure showed an average displacement value ~11.31. The double mutants L452R/T478K, L452R/E484Q and E484K/N501Y showed average displacement values ~10.88, ~7.33, and ~9.06, respectively, lower than the wild-type but higher than those in the single mutants (Appendix A).

### 3.2. Conformational Changes in RBD Double Mutants

To directly visualize the conformational changes in double-mutated RBDs, we superimposed structures between wild-type, single mutants, and double mutants. We observed that the conformational changes in each of the double mutant RBDs were substantially different from these in their corresponding single mutants (Figure 3). L452R/T478K showed conformational change at residue positions 475–482 and 518–521 (Figure 3G), whereas both L452R/E484Q and E484K/N501Y were at residue position 475–485 (Figure 3H,I). L452 in the middle and N501 at the end of the RBD–ACE2 interface of the β strand are attributed to the changes in conformations of related mutants. L452R induced conformational change at residue positions 474–485 and 517–526 (Figure 3B), whereas N501Y did so at residue positions 439–453 and 498–502 (Figure 3F). The conformational changes in double mutants comprising L452R and N501Y were very different from their corresponding single mutants.

### 3.3. Free Energy Changes in RBD Double Mutants

We used the MM/GBSA method to estimate the overall changes in free binding energy between RBD mutants and the ACE2 receptor. The high binding energy of RBD to the ACE2 receptor is largely contributed by several key residues including N501, L452, and T478, which had four, two, and two contact regions, respectively, whereas other residues, such as E484, contributed less binding energy than N501 to the ACE2 receptor (Figure 4). Comparing to the overall free binding energy of 212.5 kJ mol^−1^ between the wild-type RBD and ACE2 receptor, all single and double RBD mutants showed increased free binding energy, except for N501Y, which decreased to −204.6 kJ mol^−1^ (Table 1A). However, the changed levels between double mutants and their corresponding single mutants were at similar levels, implying that double mutants did not generate higher binding energy than the single mutants.

We also compared the energy changes in the non-mutated residues in the RBD mutants. In the wild-type RBD, a set of residues made high contributions to the overall energy between the RBD and ACE2 receptor with the top five residues of F456, F486, Q493, T500, and Y505 (Table 1B). However, the top five residues in each RBD double mutant were mostly changed from the wild-type RBD and their corresponding single mutations: L452/T478K changed to Y505, F486, N501, Q493, and T500; L452R/E484Q to F486, Q493, Y505, Y489, and F456; and E484K/N501Y to Y501, Y505, Q493, T500, and F486. N501 in the wild-type RBD contributed only −7.58 kJ mol^−1^ (9th) in the 198 RBD residues (data not shown). However, N501 became a top residue in two of the three RBD double mutants and four of the five RBD single mutants, highlighting that N501Y in the RBD double mutants and single RBD mutants played significant roles in enhancing the affinity between RBD mutants to the ACE2 receptor by enhancing the binding energy (Table 1B, Figure 4).

### 3.4. Antibody Escaping in RBD Double Mutants

The SARS-CoV-2 neutralizing antibody (NAb) C121 (PDB ID: 7K8X) is a well-known neutralizing antibody to SARS-CoV-2 through its binding to S RBD and the cell-based infectivity assay showed that E484K is resistant to C121 binding [18,19]. We used the C121 neutralizing antibody and E484K structure as the model to test the relationship between RBD double mutants and antibody resistance binding sites through structural mapping at 100 ns. Figure 5 demonstrated the independent binding sites of C121 on S protein, covering N439, N440, L455, G446, E484, and Q493, and confirmed that E484K altered the binding sites of the C121 structure. The binding sites for the double mutants L452R/T478K, L452R/E484Q, and E484K/N501Y were altered. L452R/E484Q was the most significant among the three RBD double mutants, followed by E484K/N501Y and L452R/T478K. The results indicated that the double mutants not only contributed to the increased transmission, but also the antibody escape ability of SARS-CoV-2.

## 4. Discussion

It has been proposed that higher transmissibility of new SARS-CoV-2 mutant strains can be attributed to the increased affinity of the mutated RBD to the ACE2 receptor in the host cells and that the RBD double mutations can further increase the affinity and enhance the transmissibility, as represented by the SARS-CoV-2 L452R/T478K (delta), L452R/E484Q (kappa), and E484K/N501Y (beta and gamma) strains [29,30]. By using the experimentally determined SARS-CoV-2 RBD–ACE2 structure as the reference and multiple computational tools to test the impact of the double mutations on the structure, our study provides structural-based evidence to suggest that the RBD double-mutated RBD structure can enhance the affinity of the mutated RBD to the ACE2 receptor. Together with mutations in other parts of the virus genome, they can contribute to the increased transmissibility of the SARS-CoV-2 double mutant strains.

It is interesting to note that many of the “hot spot” mutants predicted by previous studies have rarely become highly transmissible strains [31,32,33,34]. Instead, only limited single mutations were truly selected to cause outbreaks. The interesting fact is that the RBD double mutations in the newly emerging strains still inherited these single mutations at the same positions but with different combinations. This indicates that although a simulation study is powerful in detecting the structural changes caused by a mutation based solely on the physical relationship between the RBD and ACE2 receptor, in the randomly detected mutation pool, it is not a reliable approach to identify “hot spot” mutations as markers to predict the new strains with increased transmissibility, as only these with the best fitness within the mutation pool will be selected. The selection is determined by multiple factors, not only by the relationship between the RBD and ACE2 receptor. Similar situations may also exist for the newly identified B1.1.1529 (Omicron) strain, which contains over 30 mutations in the spike protein (https://www.who.int/news/item/26-11-2021-classification-of-omicron-(b.1.1.529)-SARS-CoV-2-variant-of-concern, accessed on 26 November 2021). In fact, the accumulated quantity of mutation sequences may give incorrect information for the increased transmissibility, as the number of sequences collected could be due to biased sampling, not necessarily reflecting the actual abundance of the mutant strains in the actual infected population. This is reflected by the RBD mutation sequence data in that only 4 (N501Y, L452R, E484K, Y478K) were listed among the top 7 mutation sequences, but E484Q was listed at the 29th position (Appendix A). Therefore, focusing on the “real world” mutations, either single, double, and possibly triple combinations at these fixed positions can provide more reliable evidence to monitor the newly selected strains and to use the physical evidence to explain the structural basis for these highly selected mutants with high transmissibility [35].

N501 is a particularly interesting residue. It has been determined that N501 enhances the RBD binding to ACE2 by maintaining the RBD at open conformation, and that this function can be enhanced by E484K [36]. N501 is also a mutation hot spot, as reflected by the RBD single mutation N501Y in the alpha strain [37] and RBD double mutation E484K/N501Y in beta and gamma strains. We observed that each tested RBD single or double mutation increased the free binding energy between N501 and the ACE2 receptor (Figure 4B), highlighting that N501 can be a key player in enhancing wild-type and mutated RBD binding to ACE2, not only for N501Y but also the mutated residues in other RBD locations, likely through allosteric activity [38]. For example, E484K and E484Q substantially increased the binding affinity of N501 and N501Y to the ACE2 receptor [39,40]. A similar situation was also present in RBD double mutants, and mutations outside the RBD can also contribute to the increased affinity of the RBD to the ACE2 receptor, such as D614G [41] breakthrough infection by delta strain has been observed in the vaccinated individuals [42,43,44,45,46]. Therefore, not only the mutated RBD residues, but also non-mutated RBD residues need to be considered when addressing the increased transmissibility of new mutant strains. It is interesting to note that the binding energy between single mutants (L452R, T478K, E484K, E484Q and N501Y) and double mutants (L452/E484Q, L452/T478K and E484K/N501Y) were at similar levels, indicating that double mutants cause the change in binding energy in a similar way to the single mutants.

In conclusion, our study suggests that the double-mutated RBD in SARS-CoV-2 (L452R/T487K, L452R/E484Q and E484K/N501Y) can cause changes in the RBD structure, enhance RBD binding to the ACE2 receptor, and change antibody binding sites on the RBD. These changes can contribute to the increased transmissibility of SARS-CoV-2 mutant strains with a double-mutated RBD.

## Figures and Tables

**Figure 1 viruses-14-00001-f001:**
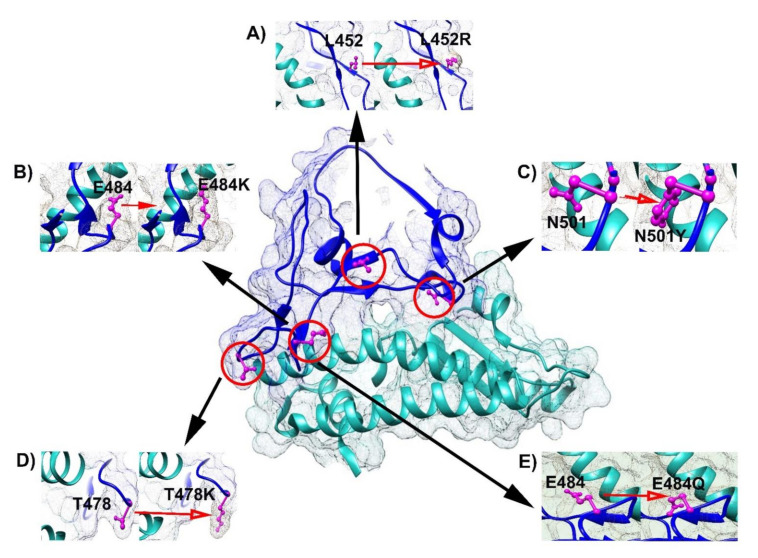
Locations of single mutated RBD and ACE2. The center shows the RBD–ACE2 complex. Blue color: RBD; grey color: ACE2. (**A**) L452 and L452R; (**B**) E484 and E484K; (**C**) N501 and N501Y; (**D**) T478 and T478K; (**E**) E484 and E484Q. Magenta color: amino acid substitutions.”.

**Figure 2 viruses-14-00001-f002:**
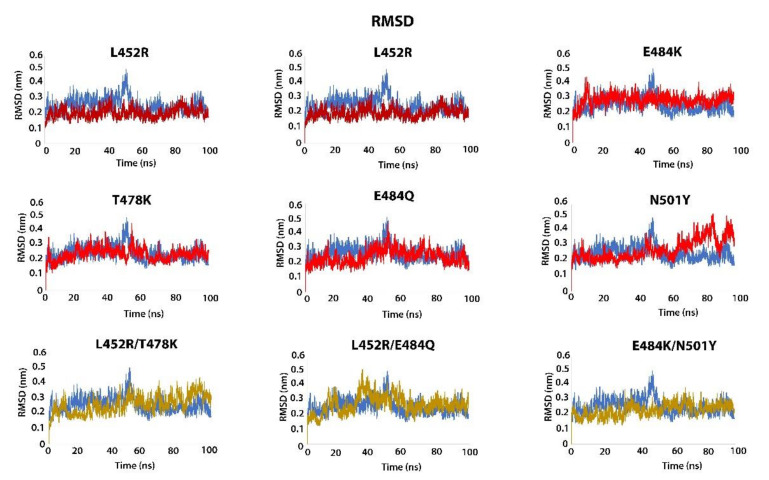
Dynamic changes of RBD structure by RMSD analysis. Blue: Wild-type, Red: single mutations (L452R, T478K, E484K, E484Y, N501Y); Yellow: double mutations (L452R/T478K, L452R/E484Q, E484K/N501Y). *x*-axis shows the simulation lasted time (100 ns); *y*-axis shows the RMSD value for the structures. The results show substantial differences between single- and double-mutated RBD structures.

**Figure 3 viruses-14-00001-f003:**
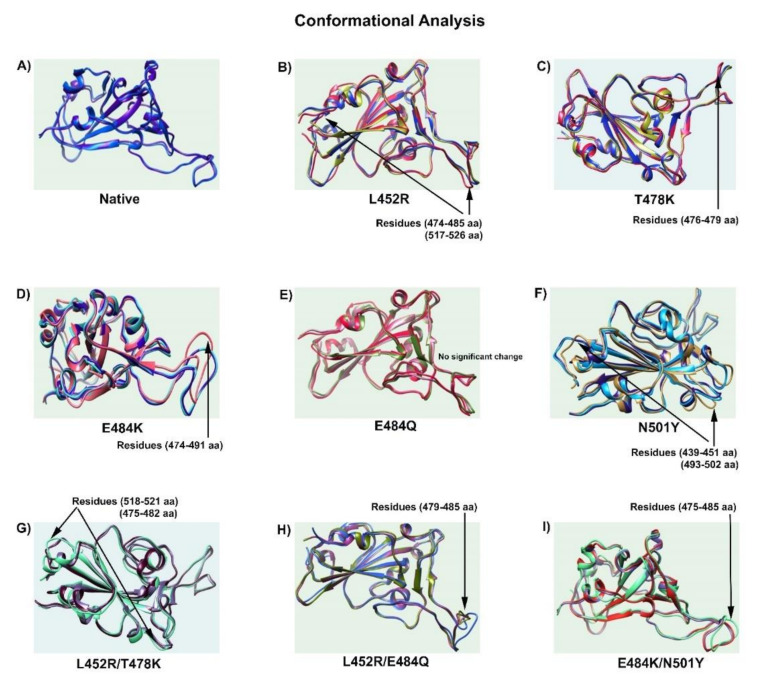
Conformational changes in mutated RBD. Mutant structures were superimposed with the wild-type RBD. Arrows show conformation changes at specific residue positions. (**A**) Native; (**B**) L452R; (**C**) T478K; (**D**) E484K; (**E**) E484Q; (**F**) N501Y; (**G**) L452R/T478K; (**H**) L452R/E484Q; (**I**) E484K/N501Y. The results show that the RBD conformational changes in double mutations were substantially different from single mutations.

**Figure 4 viruses-14-00001-f004:**
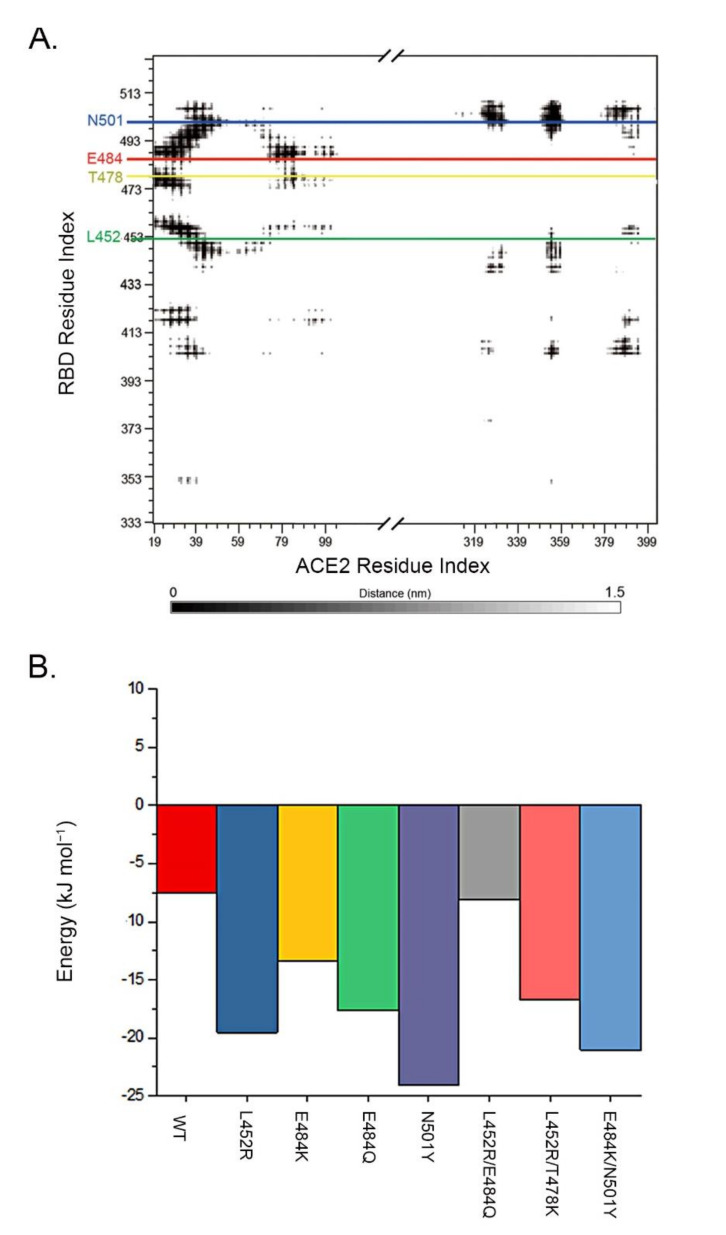
Free binding energy changes between mutated RBD and ACE2 receptor. (**A**). RBD and ACE2 residue interactions. Steps of 5 nm are represented by diminishing grey scale. The key mutation positions (L452, T478, E484, N501) in RBD are highlighted in green, yellow, red, and blue; (**B**). Binding energy by MM/GBSA for single RBD residues at position 501. It shows that although N501Y caused the highest increase in the binding energy at position 501, all single mutations (L452R, E484K, E484Q) and double mutations (L452R/E484Q, L452R/T478K, E484K/N501Y) also increased the binding energy at position 501. The red, blue, gold, green, purple, gray, pink, and light blue represented WT, L452R, E484K, E484Q, N501Y, L452R/E484Q, L452R/T478K, and E484K/N501Y, respectively.

**Figure 5 viruses-14-00001-f005:**
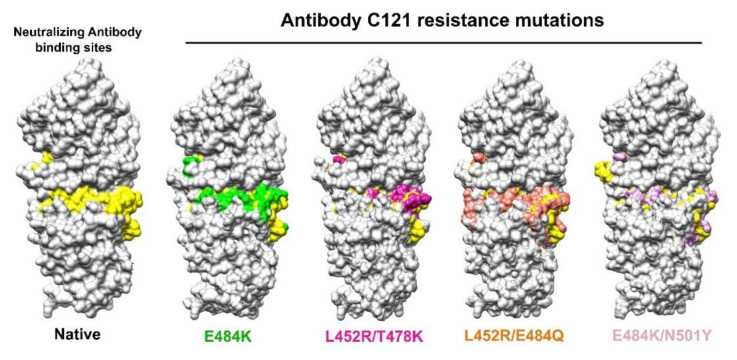
Changes of antibody binding sites in the double-mutated RBD. The binding sites (N439, N440, L455, G446, E484, and Q493) of neutralizing antibody C121 (PDB ID: 7K8X) were compared between the Native, E484K, known for escaping C121 binding, and double mutants L452R/T478K, L452R/E484Q, and E484K/N501Y. Yellow: C121 binding sites on Native RBD; Green: C121 binding sites on E484K RBD; Magenta: C121 binding sites on L452R/T478K RBD; Orange: C121 binding sites on L452R/E484Q RBD; Pink: C121 binding sites on E484K/N501Y RBD. It shows that all three double-mutated RBDs altered C121 binding sites at different degrees, as reflected by the changed yellow color in each case. L452R/E484Q showed greater overlap with yellow color than other double mutants, confirming the resistance of E484K to C121.

**Table 1 viruses-14-00001-t001:** Free binding energy changes between mutated RBD and ACE2.

Mutant	Residue	Free Binding Energy (kJ mol^−1^)
A. Overall free binding energy changes		
Wild-type		−212.5
L452R		−288.1
T478K		−229.6
E484Q		−250.2
E484K		−243.0
N501Y		−204.6
L452R/T478K		−256.1
L452R/E484Q		−252.6
E484K/N501Y		−264.5
B. Top 5 non-mutated residues with free binding energy changes		
Wild-type	Q493	−19.50
	Y505	−19.12
	F486	−14.85
	T500	−12.55
	F456	−11.41
L452R	Q498	−23.58
	Q493	−19.57
	N501	−19.54
	Y505	−18.89
	F486	−16.23
T478K	Y505	−19.15
	F486	−15.92
	Q493	−14.16
	T500	−13.62
	F456	−11.63
E484K	Y505	−21.61
	F486	−15.50
	T500	−13.73
	N501	−13.36
	Y489	−12.77
E484Q	Q493	−18.24
	N501	−17.64
	Y505	−17.16
	F486	−15.14
	F456	−12.33
N501Y	Y501	−24.05
	F486	−15.66
	Y505	−14.54
	F456	−12.23
	Q493	−11.86
L452R/T478K	Y505	−19.47
	F486	−17.49
	N501	−16.69
	Q493	−13.63
	T500	−12.99
L452R/E484Q	F486	−16.43
	Q493	−15.38
	Y505	−14.36
	Y489	−13.17
	F456	−12.56
E484K/N501Y	Y501	−21.05
	Y505	−16.95
	Q493	−16.35
	T500	−15.79
	F486	−14.90

## Data Availability

All MD simulations files are deposited at Figshare repository and can be accessed at https://figshare.com/s/5b82b4ac610e1d8403d5, Accessed 5 October 2021.

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
