# Peer review of "RBD Double Mutations of SARS-CoV-2 Strains Increase Transmissibility through Enhanced Interaction between RBD and ACE2 Receptor"

_viruses, 2021, doi:10.3390/v14010001_

Round 1

Reviewer 1 Report

The constant evolution of SARS-CoV-2 generates new mutations. Besides the well-known single mutation in receptor-binding site (RBD) of spike protein, the recent new mutation strains with “double mutation” are causing new outbreaks globally. To determine whether the interactions between ACE2 receptor and RBD of these new mutation strains are altered, several methods of molecular dynamics simulation superimposed structural comparison, free binding energy estimation, and antibody escaping were used. The results in this article demonstrated that each of the three RBD double mutants altered RBD structure and enhanced binding of the mutated RBD to ACE2 receptor.

Several minor comments:

  1. Fig. 5; Please describe why the yellow color is also shown in the mutants besides native. And, also please explain why the yellow color in E484K/N501Y is so different from those of other mutants.
  2. The altered interaction between RBD and ACE2 receptor may not be the only factor to cause the increased transmissibility of the double mutated strains. It is better to rephrase the title and lines 20-22 [and increased transmissibility of SARS-CoV-2 to host cells.].

Author Response

Reviewer 1

Question

Fig. 5; Please describe why the yellow color is also shown in the mutants besides native. And, also please explain why the yellow color in E484K/N501Y is so different from those of other mutants.

Answer

“the yellow color is also shown in the mutants besides native”:  it implies that the mutated residue(s) can only partially but not entirely change the binding area.

“why the yellow color in E484K/N501Y is so different from those of other mutants”: it implies that E484K/N501Y causes more epitope change than other double mutation.

In the revision, the following sentences are included in the Figure 5 legend to explain the issue:

Figure 5: Changes of antibody binding sites in double mutated RBD. The binding sites (N439, N440, L455, G446, E484 and Q493) of neutralizing antibody C121 (PDB ID: 7K8X) were compared between the Native, E484K known escaping C121 binding, and double mutant L452R/T478K, L452R/E484Q and E484K/N501Y. Yellow: C121 binding sites on Native RBD; Green: C121 binding sites on E484K RBD; Magenta: C121 binding sites on L452R/T478K RBD; Orange: C121 binding sites on L452R/E484Q RBD; Pink: C121 binding sites on E484K/N501Y RBD. It shows that all three-double mutated RBD altered C121 biding sites at different degree as reflected by the changed yellow color in each case.

Question

The altered interaction between RBD and ACE2 receptor may not be the only factor to cause the increased transmissibility of the double mutated strains. It is better to rephrase the title and lines 20-22 [and increased transmissibility of SARS-CoV-2 to host cells.].

Answer

Indeed, mutations out of the RBD region can also play significant roles in increasing the transmissibility of SARS-CoV-2. In fact, we indicated the fact under Discussion “mutations outside RBD can also contribute to the increased affinity of RBD to ACE2 receptor, such as D614G. Therefore, not only the mutated RBD residues but also non-mutated RBD residues needs to be considered when addressing the increased transmissibility of new mutant strains.”.

Following the comments, we have made the following changes:

The original title

“Altered interaction between RBD and ACE2 receptor contributes to increased transmissibility of SARS CoV-2 strains with RBD double mutations”

has been revised as

“Double mutations in SARS CoV-2 RBD can increase the transmissibility through enhanced interaction between RBD and ACE2 receptor”.

The original sentence between (lines 20-22) “The results demonstrated that each of the three RBD double mutants altered RBD structure, leading to enhanced binding of the mutated RBD to ACE2 receptor and increased transmissibility of SARS-CoV-2 to host cells”

has been revised as

 “The results demonstrated that each of the three RBD double mutants altered RBD structure and enhanced the binding of the mutated RBD to ACE2 receptor. Together with the mutations in other parts of the virus genome, the double mutations increase the transmissibility of SARS-CoV-2 to host cells”

Reviewer 2 Report

With this work, Sinha et al. analyzed the impact of mutations in the SARS-CoV-2 spike on RBD conformation and subsequent interaction with the receptor ACE2. The manuscript is well written and presented. The major issue with this work is that it is only computational predictions. In this regard, much of authors’ discussion is misleading as it tends to claim that the results have been experimentally validated, which is clearly overstated and not in line with the results shown here.

Specific points:

  1. Figure 1, structure of RBD and ACE2 (color) should be better indicated in the legends. What does “Table 452” mean in the legends (line 75)?
  2. Figure 2, the fact that the graph for L452R appears twice is confusing.
  3. Line 297, a space should be removed after “(…) at open conformation”.

Author Response

Reviewer 2

Question

The major issue with this work is that it is only computational predictions. In this regard, much of authors’ discussion is misleading as it tends to claim that the results have been experimentally validated, which is clearly overstated and not in line with the results shown here.

Answer

Thanks for the comments. Strictly speaking, our study is the combination of experimental and computational approaches as the structure of SARS-CoV-2 RBD - ACE2 receptor complex (PDB ID: 6M0J) was determined experimentally by X-ray crystallography at 2.45A, which was used as the standard reference in our study (Lan et al. Nature 2020, 581, 215-220). In reflecting the concerns, we have made many changes under Discussion in the revision to change the “claims” the related words with more conserved ones. Please see the marked changes in the revision.

In the revision, the following sentence is added to reflect this issue:

By using the experimentally determined SARS-CoV-2 RBD - ACE2 structure as the reference and multiple computational tools to test the impact of the double mutations on the structure,…..

Question

Figure 1, structure of RBD and ACE2 (color) should be better indicated in the legends. What does “Table 452” mean in the legends (line 75)?

Answer

“Table 452” is a typo. It should be A) L452 and L452R

Question

Figure 2, the fact that the graph for L452R appears twice is confusing.

Answer

Sorry for the confusion. This is due to the fact that L452R is present in both L452R/T478K and L452R/E484Q. Therefore, L452R is used twice for each double mutant.

Question

Line 297, a space should be removed after “(…) at open conformation”.

Answer

Removed.

Round 2

Reviewer 2 Report

The authors have addressed my concern, and the manuscript has been improved.